# A Synthetic Review of Various Dimensions of Non-Destructive Plant Stress Phenotyping

**DOI:** 10.3390/plants12081698

**Published:** 2023-04-18

**Authors:** Dapeng Ye, Libin Wu, Xiaobin Li, Tolulope Opeyemi Atoba, Wenhao Wu, Haiyong Weng

**Affiliations:** 1College of Mechanical and Electrical Engineering, Fujian Agriculture and Forestry University, Fuzhou 350002, China; ydp@fafu.edu.cn (D.Y.); LB@fafu.edu.cn (L.W.); lixiaobin592@163.com (X.L.); atobatolulope2@gmail.com (T.O.A.); foddcus@gmail.com (W.W.); 2Fujian Key Laboratory of Agricultural Information Sensing Technology, College of Mechanical and Electrical Engineering, Fujian Agriculture and Forestry University, Fuzhou 350002, China

**Keywords:** plant stress phenotyping, spectroscopy, imaging, multi-dimension, temporal, spectral, spatial

## Abstract

Non-destructive plant stress phenotyping begins with traditional one-dimensional (1D) spectroscopy, followed by two-dimensional (2D) imaging, three-dimensional (3D) or even temporal-three-dimensional (T-3D), spectral-three-dimensional (S-3D), and temporal-spectral-three-dimensional (TS-3D) phenotyping, all of which are aimed at observing subtle changes in plants under stress. However, a comprehensive review that covers all these dimensional types of phenotyping, ordered in a spatial arrangement from 1D to 3D, as well as temporal and spectral dimensions, is lacking. In this review, we look back to the development of data-acquiring techniques for various dimensions of plant stress phenotyping (1D spectroscopy, 2D imaging, 3D phenotyping), as well as their corresponding data-analyzing pipelines (mathematical analysis, machine learning, or deep learning), and look forward to the trends and challenges of high-performance multi-dimension (integrated spatial, temporal, and spectral) phenotyping demands. We hope this article can serve as a reference for implementing various dimensions of non-destructive plant stress phenotyping.

## 1. Introduction

Unfavorable factors that affect the metabolism, growth, or development of plants are known as stressors [1,2]. Plants have developed a variety of adaptive strategies to deal with environmental stressors, enabling them to survive and even thrive in unfavorable conditions [3]. These observable traits, which arise from the interaction between a plant’s genotypes and the environment, are the phenotypes we aim to obtain. Understanding these stress-induced phenotypes can aid plant breeders in developing stress-tolerant plant varieties, and can inform the development of management strategies to mitigate the effects of environmental stress on plants [3,4,5]. This is crucial for ensuring food safety and ecosystem conservation [6,7]. To accomplish this, a comprehensive understanding of plant stress phenotyping techniques is necessary [3,8].

Although plant stress phenotyping can be carried out using either destructive (biochemical analysis) or non-destructive (optical sensing) techniques [9,10], it is almost a consensus that non-invasive optical sensing technologies are highly suitable for seeing and examining plants under stress, as these methods are rapid, reliable, and repeatable [3,11]. Optical sensing utilizes optical waves (electromagnetic radiation) to interact with the object and feedback spectral characteristics. Under stress, plant physiology and morphological properties undergo a series of changes, affecting the plant’s spectral characteristics [12,13]. Specifically, when light waves strike a plant, they may be absorbed, transmitted, or reflected. In some cases, they may also be re-emitted as fluorescence. All of these can be detected as spectral characteristics [10,14]. These spectral signals can be non-destructively sensed by optical sensors. Non-destructive optical plant stress phenotyping begins with traditional one-dimensional (1D) spectroscopy, which originated mainly from remote sensing of vegetation research [15,16]. One-dimensional spectroscopy technologies used for plant phenotyping are mainly based on the reflectance or fluorescence properties of the plant, which can be further classified into (i) Visible and Infrared (Vis-IR) reflectance spectroscopy and (ii) chlorophyll fluorescence (ChlF) spectroscopy. Vis-IR spectra reflect not only morphological leaf traits (such as texture, thickness, or internal structure) but also biochemical component content [17,18]. ChlF spectroscopy can provide insights into the leaf’s photosynthesis function, particularly the capacity to withstand stressors and the degree to which the stressors have harmed the photosynthetic apparatus [19,20,21]. Vis-IR, covering 380 nm–2400 nm, especially hyperspectral spectroscopy, which provides abundant spectral information, can also be viewed as spectral-1D (S-1D), while ChlF spectroscopy is more like temperal-1D (T-1D). However, both Vis-IR and ChlF spectroscopy can only represent a limited part of the leaf.

With the development of imaging sensors, 2D imaging can provide both spatial and spectral information. Hence, 2D imaging is capable of differentiating spatial heterogeneity and also has the potential to realize high throughput. Moreover, imaging is more intuitive to human vision. Therefore, 2D imaging is now the most widely used form of plant stress phenotyping [22,23,24]. Two-dimensional imaging comprises visible, multispectral, hyperspectral, IR, thermal-IR, and ChlF imaging [22,25,26]. Similarly, 2D imaging can be divided into S-2D (visible, multispectral, hyperspectral, and IR imaging) and T-2D (ChlF imaging) in general conditions.

Projecting 3D objects onto 2D images results in the loss of some information. With the ability to differentiate plant architecture and parameterize organs, whole plants, or even population canopies, 3D measuring is crucial for a comprehensive understanding of plant stress resistance [27,28,29]. Additionally, the effect of occlusion, which is a major obstacle in 2D phenotyping methods, can be significantly diminished. In order to obtain the dynamic process of acclimation under stress, data in temporal, spectral, and spatial dimensions should be collected together. Though there remain challenges in technical issues, the development of data-acquiring and data-analyzing techniques keeps advancing to meet ever-changing needs [12,30,31].

To summarize, Figure 1 shows an overview of the data-acquisition process for non-destructive optical-based plant stress phenotyping. According to the type of data being collected, non-destructive plant stress phenotyping can be further divided into (i) 1D spectroscopy; (ii) 2D imaging phenotyping; (iii) 3D phenotyping (including T-3D and/or S-3D). However, a synthetic review that encompasses these various dimensions of phenotyping, ordered from 1D and 2D to 3D, along with spectral and temporal dimensions, has been rarely discussed. In this review, we provide an overview of the development of various types of dimensional plant stress phenotyping, from non-imaging to imaging or even videoing techniques, and their corresponding data-analyzing algorithms, from mathematical statistics to machine learning and deep learning.

Undoubtedly, there are a large number of off-the-shelf commercial phenotyping platforms equipped with various sensors (data-acquiring) and corresponding data-processing devices to extract stress phenotypes. However, they require a relatively high capital investment, and at times, may not be flexible enough to meet our specialized measuring requirements. A complete set of plant phenotyping systems mainly includes (i) the carrying platform; (ii) the data-acquiring (sensing) equipment; and (iii) the data-analyzing techniques [31,32,33]. In this context, we are not focused on the commercialized synthetic phenotyping system or the carrying platform but only on the data-acquiring and downstream data-processing techniques.

The article is divided into six sections, with the first introducing the background information and setting the context, emphasizing the importance of a synthesis review for various dimensions of phenotyping. Section 2, Section 3 and Section 4 cover phenotyping of 1D, 2D, and 3D, respectively, including data-acquiring technologies and corresponding data-analyzing methods. Section 5 discusses the advantages and disadvantages of different plant stress phenotyping dimensions, and proposes trends for multi-dimensional phenotyping, namely, combining spatial, spectral, and temporal dimensions. The conclusion summarizes various dimensions of plant stress phenotyping techniques, and emphasizes the need for an appropriate blend of these dimensions for precise and timely stress detection.

## 2. Phenotyping of 1D

One-dimensional spectroscopy technologies used for plant phenotyping can be classified into (i) Vis-IR spectroscopy and (ii) ChlF spectroscopy, as Figure 1 shows.

### 2.1. Vis-IR Reflectance Spectroscopy

Spectral reflectance characteristics of the leaf in Vis-IR regions (380–2500 nm) can offer details about the leaf’s structure, chlorophyll, water content, and other biochemical components [13,17]. The visible region (380–780 nm) provides information about spectral features of the pigments (such as chlorophyll *a* and b, carotenoids, and phytochrome) [17]. Chlorophyll *a* absorbs most at wavelengths of 430 nm and 660 nm, chlorophyll b absorbs most at wavelengths of 450 nm and 640 nm, while carotenoids absorb wavelengths in the range of 425 to 475 nm [34,35]. In addition, a sharp change in leaf reflectance between 680 and 750 nm, also termed “Red Edge”, has been measured on leaves of various species [36]. This parameter is meaningful for assessing chlorophyll status and is suitable for early stress detection. The IR region (800–2500 nm), covering NIR (Near Infrared) and SWIR (Short Wave Infrared), is associated with the measurement of overtones and combination tones of molecular vibrations, such as the O-H, N-H, C-H, and C-O covalent bonds of macromolecules in water, proteins, sugars, and cellulose [17]. Near Infrared spectroscopy is able to identify the presence of water and the physical structure of cells, as its light can penetrate deeper than visible light. With the development of hyperspectral spectroscopy, we can obtain more spectral information [37,38]. In particular, hyperspectral spectroscopy can possess a broader waveband (350 nm–2500 nm domain) with higher spectral resolution (0.1 to 1 nm), covering Ultra Violet, Visible, NIR, and SWIR wavebands [2,39]. This spectral “fingerprint” contains abundant information.

By applying appropriate spectral analysis, the changes in the reflectance signal can be extracted to characterize the plant’s phenotype and even assess plant-specific genotype responses to biotic and abiotic stress [40,41,42]. For instance, Matthew [42] discusses the use of reflectance spectroscopy for the early detection of plant physiological responses to different levels of water stress. One-dimensional (1D) spectral analysis has the advantage of providing abundant information with relatively smaller amounts of data. In addition, it is also the basis for feature extraction for images or higher-dimension phenotyping. Hence, 1D spectroscopy is still widely used for sensing plant stress conditions [43,44,45]. Nevertheless, 1D spectroscopy is mainly performed on a limited spot of leaves, so only a part of spectral information is tested to reflect the whole plant, which is certainly not enough. What is more, the environmental factor is one of the most common difficulties during outdoor data acquisition. Meteorological conditions can affect spectral reflectance characteristics and cause biases [17].

### 2.2. Chlorophyll Fluorescence (ChlF) Spectroscopy

Chlorophyll fluorescence (ChlF) can provide insights into a leaf’s photosynthesis function. More precisely, it can give information about the state of PS-II (Photosystem II: the light reaction of photosynthesis includes two photosystems, namely PS-I and PS-II, and PS-II is the first protein complex in the light reaction process) by reflecting the ability of PS-II to use the light absorbed by chlorophyll and the extent to which PS-II has been damaged under stress [14,20,46]. Chlorophyll fluorometry was developed by Kautsky [47] in the late 1960s. Since then, different ChlF testing methods have been developed, such as PAM (Pulse Amplitude Modulation) [48], OJIP (the fast polyphasic rise of the induction by continuous excitation) [49], and FRR (the Fast Repetition Rate Fluorometry) [50]. Among these, PAM still seems to be the most influential tool in ChlF and the detector can measure fluorescence regardless of ambient light interference [48]. The PAM method measures the ChlF through high-frequency modulation pulses to test the actual photosynthetic performance of the plant. Further, we can obtain a number of basic chlorophyll fluorescence parameters, including the maximum quantum yield of PS-II photochemistry (*F_v_*/*F_m_*), the effective quantum yield of PSII photochemistry (*Φ_PSII_*), and the non-photochemical quenching (*NPQ*) of ChlF [51,52]. There are also alternative methods, namely, OJIP-test or fast (prompt) ChlF. OJIP means a fast (often fewer than 1 s) increasing phase of chlorophyll *a* fluorescence induced by constant light excitation. O is the origin (the minimum fluorescence), J and I are in the intermediate levels, and P is the peak [47]. One of the most typical testing methods is using a PEA (Photosynthetic Efficiency Analyzer) instrument [47]. It can measure the fast fluorescence induction kinetics, which reflects the state of the electron transport chain in PS-II. Such analyses offer detailed insights into the functional and structural status of PS-II reaction centers and antennae, as well as the donor and acceptor sides [53,54]

Although fluorescence measurements can give insights into the ability of plant stress tolerance by analyzing photosynthetic capacity, it is a limited approach. For instance, ChlF spectroscopy can only measure a small area/proportion of the leaf, so one has to measure many leaves to represent the whole plant in a short time, as the parameters can change over time. In addition, not all stress may manifest itself in certain leaves. However, as K. Maxwell et al. [14,20] suggested, combining it with other techniques is the most powerful and elegant application of fluorescence.

### 2.3. Data Analysis of the 1D Spectral Curves

Prior to the advent of machine learning, most data processing tasks were typically undertaken using mathematical and statistical software packages, such as SPSS, Microsoft Excel, MATLAB, or other specialized software kits.

#### 2.3.1. Vis-IR Reflectance Spectral Curve

The reflectance and absorption of different wavelengths of light are determined by the leaf’s cellular structure, chlorophyll, phytochrome, water content, and other biochemical components [17,55]. To extract meaningful information from the spectral curves, mathematical and statistical methods are employed. The data processing pipeline comprises three essential steps: preprocessing, calibration, and validation (prediction) steps, as Figure 2 shows.

Vegetation indices (VIs), derived from plant spectral reflectance characteristics, are widely used to assess plant status [40,56]. Many empirical and semi-empirical spectral indices have been derived from the leaf reflectance spectra and proved to be related to plant physiological status [56,57,58]. For instance, the well-known NDVI (Normalized Difference Vegetation Index) can measure the chlorophyll absorption in the red spectrum relative to the scattering by the cellular structure, and has been used to monitor stress from proximal or remote sensing images [57,59]. Additional VIs, such as those used for growth monitoring, crop yield estimation, and plant distribution, can also be extracted. VIs can be grouped into two categories: Plant Activity and Plant Productivity VIs. Plant Activity VIs (e.g., LAI: Leaf Area Index, NDWI: Normalized Difference Water Index, SAVI: Soil Adjusted Vegetation Index, EVI: Enhanced Vegetation Index) are suited to estimating the current physical state of plants, while Plant Productivity VIs (e.g., CHI: Chlorophyll Index, NDRE: Normalized Difference Red Edge, NIRV: Near Infrared of Vegetation) provide information on yields and biochemical states [60,61].

#### 2.3.2. Chlorophyll Fluorescence (ChlF) Kinetic Curve

From a typical chlorophyll fluorescence kinetic curve, one can obtain the photosynthetic function parameters of the leaf. Light absorbed by chlorophyll molecules functions in three processes: (1) driving photosynthesis (photochemistry quenching); (2) being re-emitted as heat (non-photochemical quenching); or (3) being re-emitted as light (fluorescence) [14]. The total amount of energy involved in these three processes follows the law of energy conservation. Therefore, by measuring the amount of fluorescence emission, we can also obtain information about the other two processes. In other words, if the amount of fluorescence can be measured, then the photochemical and non-photochemical parameters of the sample can be deduced, which in turn allows for the estimation of the sample’s photosynthetic ability [14]. Figure 3 depicts a stylized fluorescence trace experiment on leaves to measure ChlF parameters. Here, the parameters of chlorophyll fluorescence are divided into two groups: preliminary parameters (in the order of testing steps, as shown in Figure 3) and deduced parameters (obtained from preliminary parameters), as listed in Table 1.

To date, a variety of ChlF parameters have been calculated to be used as stress indicators [21,63]. Table 1 shows some commonly used ones deduced from the PAM test [20,49]. Alternatively, parameters from the fast ChlF test, such as OJIP, can provide different parameters representing different photosynthetic statues of chlorophyll *a*. For a more comprehensive list, please refer to [47,52,64,65]. All these parameters often can either be directly obtained in the accompanying software or be calculated using other typical mathematical methods.

## 3. Phenotyping of 2D

### 3.1. Visible Imaging

Visible-band imaging captures visible light and records images through sensitive materials such as charge-coupled devices (CCD) or Complementary Metal Oxide Semiconductors (CMOS) [25]. For instance, a mono-spectrum camera means imaging in a single color or another single visible spectrum with varying grayscale. RGB cameras represent three spectral bands red (about 600 nm), green (about 550 nm), and blue (about 450 nm) [22,25]. Analyzing visible-band imaging can provide information about morphologic and geometric properties, pigment distribution, and stress analysis [66,67,68]. This is similar to visible spectroscopy.

For example, Enders [68] provides a method to classify cold-stress responses of inbred maize seedlings using RGB imaging. Tackenberg [67] utilized RGB images for measuring biomass, including above-ground fresh biomass and dry matter content. These methods can be useful for high-throughput plant phenotyping. However, visible imaging has the drawback of providing only visual information, with limited color and gray values in the visual spectral bands. Machine vision, unlike human vision, is capable of sensing invisible light such as UV or IR light wavebands, which can provide valuable information for assessing the plant’s stress state. However, automatic downstream image processing to fit human vision may overlook crucial details [22].

### 3.2. Multispectral and Hyperspectral Imaging

Multispectral and hyperspectral imaging techniques can cover the visible (400–700 nm), near-infrared (700–1100 nm), and shortwave infrared (1100–2500 nm) spectral regions [69,70]. These wavelength bands are similar to those used in Vis-NIR spectroscopy. The visible region reflects photosynthetic pigment information; NIR and SWIR regions show water content and nitrogen information; SWIR can be used to estimate the amounts of minerals, hemicellulose, protein, and phosphorus in plant materials [2,71]. In other words, multispectral and hyperspectral imaging can be viewed as an upgraded version of non-imaging Vis-IR spectroscopy, but they can capture both spectral and spatial (2D) information. Spectral imaging achieves this by integrating imaging and spectroscopy, thus simultaneously measuring spectral and spatial information. Hyperspectral imaging can be realized through four basic techniques: point scanning, line scanning, area scanning, and snapshot, listed in the order of their development. As of now, the line scanning (or push broom) movement of the sensor is the most typical imaging system utilized for close-range hyperspectral image collection [2,71]. Multispectral imaging is similar to hyperspectral imaging but with sparse wavelength information.

For instance, Anika [72] utilized high-throughput hyperspectral imaging to detect cadmium stress in crops, and a machine learning model was developed to automatically classify the crops based on their spectral features. However, compared to RGB imaging, the acquisition of multispectral and hyperspectral imaging can be more complex and slower. Additionally, hyperspectral imaging is more susceptible to the effects of illumination and environmental factors, and greatly relies on accurate data-analysis techniques and reliable sensing systems [71,73]. However, ongoing technological advancements would make multispectral and hyperspectral imaging more high-performing and lightweight [74,75].

### 3.3. Chlorophyll Fluorescence (ChlF) Imaging

Compared to point measurement ChlF spectroscopy, ChlF imaging can detect spatiotemporal heterogeneity over the entire leaf surface [19]. One ChlF imaging system typically includes an excitation source, which can be either natural solar light or artificial UV light (with a wavelength ranging from 340 to 360 nm), and a sensitive camera (such as CCD or CMOS) that records the re-emitted fluorescence light [76,77]. As ChlF imaging can detect spatial heterogeneity, the re-emitted fluorescence light comes not only from chlorophyll, but also from epidermal cell walls and leaf veins. [20,78]. To be more precise, when excited by UV radiation, plants exhibit fluorescent light from two different wavelength bands: (i) the Red to Far-Red region, which is mainly related to chlorophyll *a*; (ii) the Blue to Green region, which is primarily emitted from the epidermis cell walls and the leaf veins by compounds such as ferulic acid [14,64].

Specifically, the Blue/Red and Blue/Far-Red fluorescence ratios are effective indicators of early plant stress. The former is related to plant structural components, while the latter is related to changes in the photosynthetic apparatus. By analyzing these ratios, researchers can identify early signs of stress before visible changes occur [21,79,80]. ChlF imaging offers advantages in providing information on spatial heterogeneity and temporal changes, making it a useful T-2D phenotyping tool. However, fluorescence parameters are not stress-specific and can be affected by various environmental conditions [81,82,83]. That is to say, many conditions would affect photosynthetic machinery, and so would fluorescence parameters. To ensure accuracy, combining ChlF imaging with other sensing technologies or developing better hardware and data-processing techniques is necessary, especially in uncontrolled field conditions.

### 3.4. Infrared (IR) Imaging

The IR region can be divided into two types: reflected-IR (NIR and SWIR: 0.7–3.0 µm) and emitted-IR (thermal-IR: 3.0–100 µm) [84]. IR imaging primarily refers to reflected-IR imaging, which covers the NIR and SWIR regions and is similar to multispectral or hyperspectral imaging. This imaging can measure various biochemicals such as water content, mesophyll cell structure, nitrogen protein, and cellulose [26,71,75]. IR imaging is often combined with RGB imaging to detect physiological status and screen morphological traits under stress.

For example, Anna Kicherer [85] utilized RGB and NIR imaging to analyze grapevines under drought stress and applied machine learning to identify patterns associated with drought tolerance. The study showed that RGB and NIR imaging can accurately differentiate between drought-tolerant and drought-sensitive grapevine varieties. The utility of IR imaging in plant stress phenotyping is valuable. However, IR fails to offer robust data on chemical composition. Furthermore, the reflectance pattern in the NIR region could be influenced by leaf thickness, growth condition, and canopy architecture. So, IR imaging is often combined with other sensing techniques [86].

### 3.5. Thermal-IR Imaging

Thermal-IR imaging, also called thermography, differs from other optical imaging techniques as it measures the emitted infrared radiation from the surface of samples rather than reflected radiation. [84]. Generally, any object with a temperature above Absolute Zero (−273.15 °C) will emit thermal-IR radiation [10]. Then, these radiation data are demonstrated as thermal images by a false-color temperature gradient [87]. In other words, thermal-IR imaging allows the visualization of temperature differences on the surface of plants caused by stress [88]. For instance, adjustments in the water status of a plant under stress lead to changes in leaf transpiration and gas conductance. These associated changes result in temperature differences and thus can be instantly and remotely sensed by thermal-IR imaging [88].

With a continuous increase in resolution (0.01 °C) [89], thermal-IR imaging is a promising approach for the detection of subtle changes in plants upon stress [84,90,91]. For example, in one study, Martínez [90] presents a methodology for the early detection of fungal infections by measuring the temperature changes in grapes using infrared thermography. However, thermal-IR imaging is vulnerable to environmental changes, such as meteorological conditions and the crosstalk of other emitted and reflected thermal radiation sources. For this reason, calibration and ground data collection are necessary. Meanwhile, data processing needs to be carried out for correct temperature retrieval [84]. In addition, just like fluorescence imaging, thermal-IR imaging lacks stress specificity since it is unable to distinguish between different stresses; thus, additional tests will be necessary to identify results.

### 3.6. Image Processing of 2D Phenotyping

Two-dimensional imaging provides more spatial information than one-dimensional spectroscopy, but also generates more data to analyze, making it challenging to process using traditional mathematical or statistical methods. Image processing techniques are then used to overcome this challenge. Image processing is a type of computer technology that processes, analyzes, and extracts useful information from images. In machine vision, a digital mono-spectrum image is represented as a matrix of definite size with different pixels having values varying from 0 to 255. An RGB image consists of three matrices, while a multispectral or hyperspectral image has more matrices, or in other words, more spectral channels, as Figure 1 shows. Machine learning algorithms are typically used in image processing to analyze these matrices and obtain relevant information.

Machine learning means the ability to learn without being explicitly programmed. It is more precisely definitized by Tom Mitchell [92] in the classic 1997 textbook: “The field of machine learning is concerned with the question of how to construct computer programs that automatically improve with experience.” That is why ML provides an excellent solution for image processing. To date, various ML techniques are blossoming. That is to say, ML is not a specific algorithm but a general term for many algorithms.

Deep learning is one branch of machine learning, or more specifically, the development results of artificial neural networks (or neural networks for brief), which attempt to simulate the behavior of the human brain [93,94,95]. It is the number of layers that sets a deep learning algorithm apart from a single neural network. Multi-layer neural networks in deep learning can automatically achieve feature extraction, unlike handcraft feature extraction in traditional machine learning. Deep learning is now widely used in image processing, in particular [95,96,97].

To make a distinction, machine learning in image processing can be further divided into traditional machine learning (TML) and deep learning (DL) as they possess different data-analysis pipelines, as Figure 4 shows. Compared to TML, DL techniques have the advantage of automated feature extraction.

As for the deployment of DL models for plant stress phenotyping, three strategies are presented, ranging from directly using pre-trained models to designing a custom model from scratch.

(1) Off-the-shelf models. If the tasks and datasets are similar, then the trained models can be directly used. Some sharing pre-trained deep learning models can be referred to, such as the plant leaves disease diagnosing model, which can be searched for on public data-sharing websites such as GitHub. However, this is rarely the case.

(2) Transfer learning. If the tasks and datasets are slightly different, although they cannot be used directly, the parameters can be changed or some changes to the architecture can be made to fit our tasks. The term transfer refers to the fact that a major portion of the model weight and trainable parameters are frozen, facilitating the use of previously learned features [98,99]. The changes usually occur at the end of the network concerning the classification number or type of outputs (discrete or continuous) [100,101].

(3) Build from scratch. If the tasks and datasets are inapplicable, then the learning model must be built from scratch. A deep learning model built from scratch requires enormous amounts of data for training and a network customized to the data. The major drawback is data acquiring and preprocessing [102,103]. The customized model architecture should be designed based on the dataset, GPU, and the task of interest [104]. However, these can add an advantage to the model’s robustness to the new datasets [93,105].

To summarize the above three methods, “off the shelf“ and “build from scratch” are two relatively extreme cases. “Transfer learning” is the most common condition, without requiring a large amount of training data, but with less training time, lower computational requirements, and appreciable performance improvement [104]. However, transfer learning for a given deep learning model can only be used for similar tasks, not for radically different ones. Moreover, at present, the availability of large-scale public datasets and pre-trained models is not exhaustive or extensive; therefore, building a model from scratch and further research are often necessary.

## 4. Phenotyping of 3D

Three-dimensional phenotyping can be achieved using various techniques, but almost all are reconstruction models generated by computers [106]. According to the information being measured, 3D phenotyping can be further divided into surface and internal (anatomical) traits. (i) Surface traits can be obtained using laser scanners (such as Light Detection and Ranging) and photogrammetry (such as stereo cameras) techniques. Laser scanners are used to obtain the spatial 3D “point clouds” through scanning, while photogrammetry uses stereovision 2D images to reconstruct 3D models [107,108,109]. (ii) Internal traits are often used to detect undersurface information unseen in ordinary ways. However, they can be determined through X-ray CT (computed tomography) [110], MRI (magnetic resonance imaging) [111], or PET (Positron Emission Tomography) [112].

As for plant stress phenotyping, the surface traits of 3D models may not be fine enough to identify nuanced changes in plants under stress. Due to limitations in point cloud density, surfaces are often incomplete or sparse. For this reason, multi-view 2D imaging is often better suited for capturing surface information. However, with the development of Light Detection and Ranging (LiDAR), high-resolution 3D models are becoming increasingly promising. In the following, we describe some commonly used methods for capturing 3D features, including LiDAR for surface features, and X-ray CT, MRI, and PET for internal characteristics.

### 4.1. Light Detection and Ranging (LiDAR)

LiDAR uses pulsed lasers to build the “point cloud”. It includes a laser light source (mainly NIR of 950 nm or 1550 nm) and a receiving system. LiDAR actively emits laser light signals, using the time of flight between the source and the target to calculate the distance. Then, many light signals can create the “point cloud” to describe the 3D surface structure [113,114,115,116].

The LiDAR laser can partially penetrate the vegetation canopy and is not vulnerable to natural sunlight [117], so LiDAR sensors have been mounted on the ground and aerial phenotyping platforms to measure various phenotypic traits [114,118,119]. Although LiDAR can be used to monitor the 3D surfaces of plants from one meter up to thousands of meters [113,119], there remain some disadvantages, such as matching errors caused by illumination and shadowing, incomplete reconstruction data caused by occlusion, and tradeoffs between accuracy and efficiency [113,117]. That is the reason why LiDAR has low accuracy when performing large-scale scanning. In the future, the density of the achievable 3D point cloud needs to be increased to better describe 3D plant structures.

### 4.2. X-ray Computed Tomography (CT)

X-rays are a form of electromagnetic radiation with a wavelength ranging from 0.01 to 10 nm, and extremely high frequencies ranging from 3 × 10^16^ Hz to 3 × 10^19^ Hz. They can penetrate optically opaque materials. When the transmitted X-ray is recorded by CCD cameras or other sensitive materials, then X-ray images are produced. In short, X-ray imaging produces a transmittance image [106,120] showing discontinuities (due to different attenuation coefficients) in the material, which is quite different from typical Vis-NIR reflectance images.

Standard X-ray imaging can only produce 2D images, from which one can differentiate between different tissues, and cannot be used to acquire 3D structures [110,121]. Admirably, with the development of computed tomography (CT) techniques, now X-ray CT can provide detailed cross-sectional images of internal organs by rotating scanning [110,122]. Contrary to X-ray imaging which compresses a 3D object into a plane, X-ray CT reconstructs the 3D structure with those cross-section images. However, performing X-ray CT requires that attention is paid to the dose. In addition, X-ray CT can show spatial information but it is incompatible with metabolite analysis.

### 4.3. Magnetic Resonance Imaging (MRI)

MRI is based on the magnetic momentum nucleus (such as ^1^ H, ^13^ C, ^14^ N, ^15^ N, ^31^ P, and their bounds), which are highly abundant in living tissues [123]. Their magnetic momentum can be manipulated using strong magnetic fields and radio frequency. Due to the difference in absorption resonance frequencies [124], which can be detected to differentiate their content and generate images of the internal structure [111,123,125], the different contents are translated into various shades of grey.

It is possible using MRI to recognize internal tissues. That is why MRI can be used to visualize internal structures and metabolites; therefore, it has the potential to monitor physiological processes occurring in vivo [125,126]. MRI has been used to detect underground root internal structures or stem water transportation [127,128,129]. Since MRI necessitates large electro-magnets (commonly between 0.2 and 7.0 T), it is hard to operate directly in the field [130,131]. MRI also has the limitation that data acquisition takes a long time and has a high cost. However, there are several transportable MRI devices, such as NMR-MOUSE (mobile universal surface explorer) [124] and cut-open force-free NMR (NMR-CUFF) [132]. While their resolution is limited, this restriction could be overcome through ongoing technical improvements.

### 4.4. Positron Emission Tomography (PET)

The Positron Emission Tomography (PET) process includes: first, one of the atoms of the object compounds is substituted with a radionuclide without changing the host chemical property; second, the emitted positrons annihilate the electrons in the plant tissues, producing a distinct external signal consisting of two almost collinear gamma rays; finally, gamma rays can be detected to show the density of the object compounds [133,134]. Plant PET can provide a quantitative analysis of the dynamic function of stressed plants in a 3D view [112,135]. Alternatively, 3D structural tomography can be used to trace the changes between different tissues and offer a quantitative measurement of the transport and allocation of metabolites in plants under stress.

For example, the author in [135] notes that PET imaging has been successfully used in investigating the dynamic transport of nutrients, phytohormones, and photoassimilates. Thus, PET is a promising tool for 3D functional imaging, enabling our study of the complex interactions between plants and their environment [112,135]. However, this quantitative measurement within the plant needs a higher spatial resolution, which is challenging for accessible PET. The combination of PET and MRI is expected to be a powerful tool for understanding stress responses [134,135].

### 4.5. Data Processing of 3D Phenotyping

Tridimensional measuring enables the geometry information of the plant and individual organs to be gathered. As described earlier, 3D information includes surface traits and internal structure, so the data processing methods vary. Internal structure construction is mainly undertaken in the accompanying specialized software. The surface traits, according to the procedure of data processing, can be divided into (i) primary traits, i.e., whole plant level (such as height, width, volumetric measures); (ii) and derived traits, i.e., organ level (such as the exact leaf area, stem length, and branch number) [109,136,137]. Primary traits can be obtained using the complete plant point cloud analysis, while the derived features require previous segmentation of plant organs, and information is then derived through the organs. Figure 5 shows a stylized processing pipeline for 3D data processing coming from a common point cloud.

Using routines from standard data processing software libraries such as MATLAB [138], OpenCV [139,140], or the Point Cloud Library [141,142], primary traits can be extracted. For example, after cutting the point cloud to the region of interest and performing a data cleaning step, non-complex parameters such as height and width can be derived [143]. Machine learning approaches can then be employed for further processing, such as segmenting plant organs such as leaves, stems, and flowers [117,144,145,146]. For instance, Li [116] used a U-Net architecture, which is a popular CNN architecture for image segmentation tasks, and modified the U-Net mode to take the point cloud data as input, then output a segmentation mask that identifies the different plant organs. Finally, the corresponding traits of interest can be derived.

## 5. Discussion

Overall, plant phenotyping sensing methods have evolved from 1D spectroscopy to 2D imaging and then to 3D phenotyping, and even to T-3D and S-3D. Related data types vary from the 1D spectral curve and 2D images, to 3D models. Consequently, analyzing methods vary from mathematical and statistical methods to machine learning and deep learning algorithms, which are summarized in Table 2.

Plant response to stress is a dynamic equilibrium process accompanied by a series of morphological, physiological, and biochemical changes in different organs [147,148]. So, obtaining data at a specific time or spectrum has constraints. In order to monitor the dynamic process of acclimation under stress, a necessary complement is to combine temporal, spatial, and spectral information. Temporal and spectral information integrated with spatial architecture can be used to track changes in the growth and movement of whole plants or particular organs.

One-dimensional spectroscopy comprises Vis-IR and ChlF spectroscopy, which can be attributed to spectral-1D and temporal-1D, respectively. Two-dimensional imaging can be divided into spectral-2D (visible imaging, multispectral and hyperspectral imaging, and IR imaging) and temporal-2D (ChlF imaging) phenotyping. Three-dimensional models mainly focus on spatial architecture. To realize spectral-3D, there may remain technical obstacles; however, this approach also can be seen in some research using methods such as multispectral image registration [149,150], or the data fusion of LiDAR data with multispectral imaging [136]. By repeating measurements and/or analysis methods over time series, the temporal dimension information can be derived [151,152]. For example, MRI and/or PET can also be viewed as temporal-3D phenotyping [134,153,154]. MRI scanning allows the measurement of both the distribution and dynamics of water and other metabolites [124,125]. PET is a time-dynamical acquisition approach used for measuring transport fluxes [112,135]. Thus, spectral-3D or temporal-spectral-3D imaging systems can provide more spectral bands and temporal information. In this regard, multi-dimension combines (i) spatial information (from the 1D spot and 2D plane, to 3D stereoscopy); (ii) spectral information (multispectral, hyperspectral, IR, thermal-IR); and (iii) temporal information (temporal evolution of phenotyping variables).

In summary, each has its advantages and disadvantages. One-dimensional spectroscopy may gradually be less used, because of its limited spatial information. However, 1D spectral analysis is the basis and reference for higher-dimension image processing, and it requires less data storage while being rich in spectral information. Two-dimensional imaging technologies are the mainstream, thanks to advancements in imaging sensors and computational devices, and the availability of a wide range of processing algorithms. Three-dimensional imaging and multi-dimension approaches are important contemporary and future trends, capable of providing an excellent solution for comprehensive stress identification and prediction, although collecting various images and processing vast amounts of data are practical issues. Supported by the development of high-resolution compact portable data-acquiring devices, and high-performance computational data-processing algorithms, multi-dimension phenotyping will drive new research in agriculture.

**Table 2 plants-12-01698-t002:** Data-acquiring and corresponding data-processing methods of various dimensions of plant stress phenotyping.

Dimensions	Data-Acquiring Methods	Data Types	Data Analyzing Methods/Tools	References
**1D Phenotyping**	Vis-IR reflectance spectroscopy	S-1D	1. Chemometric methods (PCA or PLS regression), statistical methods, and tools such as SPSS, R language, etc. 2. For devices that can calculate automatically, typical mathematical methods are then used to further process the data.	[45,55,62,155,156]
	ChlF spectroscopy	1D, T-1D		
**2D Phenotyping**	Visible imaging	2D, S-2D T-2D	1. Image preprocessing, segmentation algorithms (watershed algorithm and color segmentation methods), and image conversion (wavelet analysis). 2. Image processing tools, such as Plant CV (https://plantcv.danforthcenter.org, accessed on 20 March 2023), IAP (http://iap.ipkgatersleben.de, accessed on 20 March 2023), and Image J (http://imagej.nih.gov/ij, accessed on 20 March 2023), and, etc. 3. Machine learning and/deep learning for the identification, classification, quantification, and prediction of stress phenotypes, such as support vector machine (SVM), Random Forest, Gaussian processes (GP), CNN and LSTM, etc. 4. Specialized corresponding data analysis software, such as Envi, Evince, SpecSight, as well as other proposed in-house image processing software solutions.	[22,66,95,157,158,159]
	Multispectral imaging	2D, S-2D, T-2D		
	Hyperspectral imaging	2D, S-2D, T-2D		
	ChlF imaging	2D, T-2D		
	IR imaging	2D, S-2D, T-2D		
	Thermal-IR imaging	2D, T-2D		
**3D Phenotyping**	LiDAR	3D	1. Specialized 3D software solutions for showing 3D models, as well as with algorithms such as SFM (structure from motion), voxel-based volume carving, and Stereo correspondence algorithms. 2. Dimensionality reduction is commonly used for image processing and feature extraction, while machine learning and/or deep learning are used for data analysis. 3. To capture temporal information, optical flow-based tracking and adaptive hierarchical segmentation methods can be utilized.	[143,160]
	X-ray CT MRI PET Multispectral LiDAR	3D		
		3D, T-3D		
		3D, T-3D		
		3D, S-3D		

Visible-IR: Visible Infrared; Chlorophyll Fluorescence: ChlF; CT: Computed Tomography; MRI: Magnetic Resonance Imaging; PET: Positron Emission Tomography; PCA: principal component analysis; PLS: partial least squares; S: Spectral; T: Temporal.

## 6. Conclusions

In this article, we provide an overview of the various spectroscopic and imaging techniques across spatial-temporal-spectral dimensions, which have been used for data acquisition in plant phenotyping, as well as the corresponding data processing methods. Overall, 1D spectroscopy can provide spectral information of certain spots in the leaf, 2D image-based phenotyping can provide both spatial and spectral information of the plant in plane vision, and 3D image-based phenotyping can provide stereoscopic features of the plant. In comparison, temporal-3D and/or spectral-3D can provide additional dimension insights into the morphology/anatomy of opaque samples, thus allowing an assessment of a range of physiological, morphological, and biochemical parameters. In short, it is claimed that incorporating appropriate dimensions in plant stress phenotyping can improve the accuracy and efficiency of stress detection. Hence, multi-dimension approaches that combine spatial, spectral, and temporal dimensions can be helpful, especially for complex plant stress identification/prediction.

However, multi-dimension phenotyping would undoubtedly cause the accumulation of big data. In addition, data vary in their spatial, spectral, and temporal types. Additionally, data fusion from different optical sources would also add complexity. Therefore, analyzing such heterogeneous and enormous amounts of data is rather challenging. Thus, more research is needed to handle the complexities of multi-dimensional phenotyping data. This includes the development of high-resolution sensing systems, high-performance computational graphics-processing technologies, and robust analysis pipelines. Furthermore, the integration of big data and AI-driven smart agriculture makes it increasingly feasible to achieve real-time, multi-dimensional monitoring of plants. This can aid in identifying plant stress, making predictions and recommendations for management, and accelerating breeding programs.

## Figures and Tables

**Figure 1 plants-12-01698-f001:**
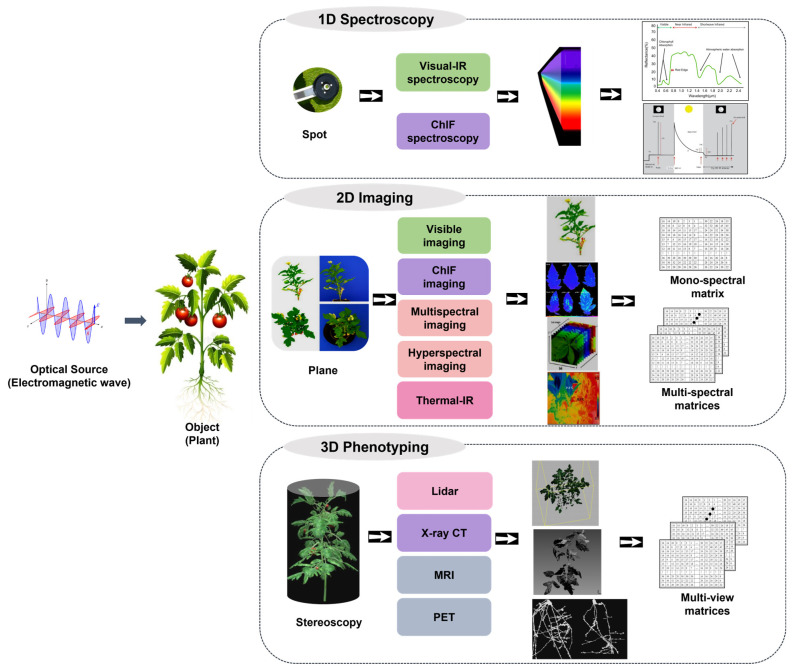
Data-acquisition process for non-destructive optical-based plant stress phenotyping, involving 1D spectroscopy, 2D imaging, and 3D phenotyping. One-dimensional spectroscopy techniques include Visible-IR and ChlF spectroscopy; two-dimensional imaging techniques include visible, multispectral, hyperspectral, IR, and thermal-IR imaging; three-dimensional phenotyping techniques include LiDAR, X-ray CT, MRI, and PET. The corresponding acquired raw data and data type are also shown in this picture.

**Figure 2 plants-12-01698-f002:**
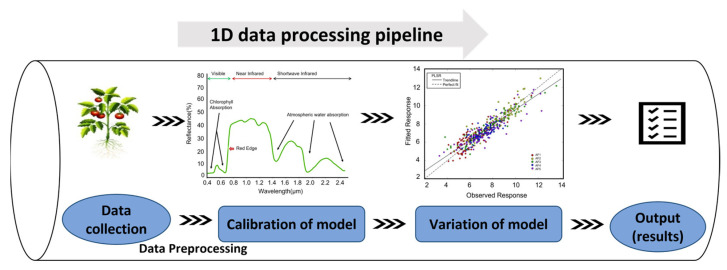
One-dimensional reflectance spectral data processing pipeline. After data collection, preprocessing is needed to remove the influence of irrelevant information and background noise in the results. Calibration of the model is used to find the correlation between the sample’s properties and absorbance, perform the fitness test of the model, and connect the attributes of samples with the preprocessed measured spectra. Validation of the model is used to predict the spectral signal of unknown samples based on the calibrated model and evaluate the model’s accuracy.

**Figure 3 plants-12-01698-f003:**
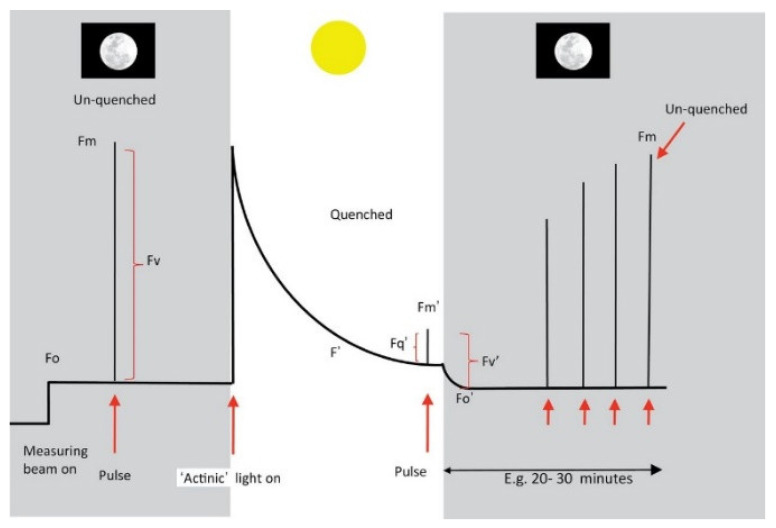
A typical chlorophyll fluorescence kinetic curve used to measure the leaf’s photochemical and non-photochemical parameters. A measuring beam means light that is too low to induce photosynthesis but high enough to elicit chlorophyll fluorescence. Actinic light means light is fit for photosynthetic function. Pulse means saturating flash that can transiently close all PS-II reaction centers, but the flash is short enough, so no increase in non-photochemical quenching occurs. (Reprinted with permission from Ref. [62]).

**Figure 4 plants-12-01698-f004:**
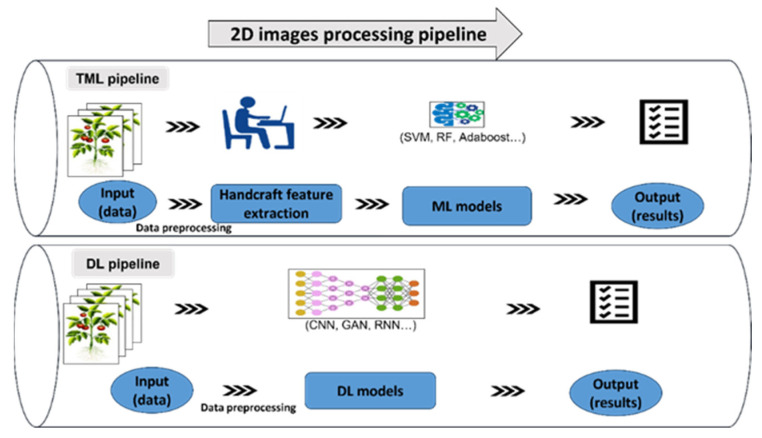
Two -dimensional image processing pipeline based on traditional machine learning (TML) and deep learning (DL). TML processing flow includes: data preprocessing, feature extraction, choosing the ML model, training the model, and finally obtaining the well-trained model (satisfying the accuracy). DL processing flow includes: preparing datasets and data preprocessing, choosing the DL model, training the model, and finally obtaining the well-trained model.

**Figure 5 plants-12-01698-f005:**
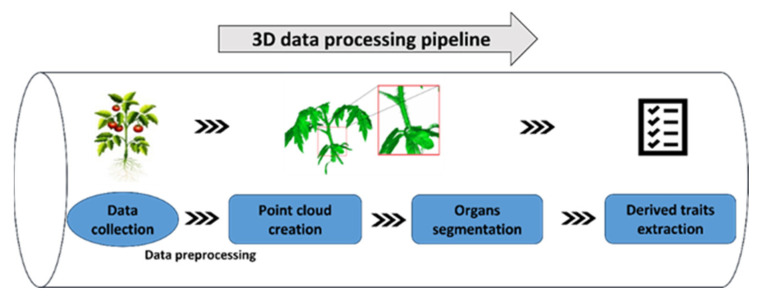
Three-dimensional phenotyping data processing pipeline. Data processing flow includes: data collection, data preprocessing, point cloud creation and organ segmentation, and derived traits extraction.

**Table 1 plants-12-01698-t001:** ChlF parameters measured from the typical chlorophyll fluorescence kinetic curve.

	Parameter	Measurement and Calculation	Description
Preliminary parameters	*F_o_*	Switch on the measuring light, and get the parameter *F_o_.*	Minimal fluorescence of chlorophyll *a* in dark-adapted leaves, indicating the baseline fluorescence of the sample.
	*F_m_*	Offer a pulse, then induces *F_m_*.	Maximal fluorescence of chlorophyll *a* in dark-adapted leaves.
	*F’*	Switch on actinic light, followed by an initial rise in fluorescence. Then fluorescence quenches due to the increasing competition with photochemical and non-photochemical events. This state is also named the light-adapted state.	It represents the chlorophyll fluorescence yield in the light-adapted state in the presence of actinic light.
	*F_m_′*	Offer a pulse to the light-adapted state, and get the parameter *F_m_′*.	Maximal fluorescence of chlorophyll *a* in light-adapted leaves.
	*F_o_′*	Switch off the actinic light and measure immediately, recording *F_o_′.* However, the accurate measurement is complex; an alternative approach is to calculate *F_o_′.*	Minimal fluorescence of chlorophyll *a* in light-adapted leaves.
Deduced parameters	*F_v_*	*F_v_ = F_m_ − F_o_*, the difference between *Fm* and *Fo* is the variable fluorescence *F_v_*.	It is related to the maximum quantum yield of PS-II, reflecting the amount of chlorophyll molecules that are in the open reaction centers and actively.
	*F_v_/F_m_*	*Fv*/*Fm = (F_m_ − Fo)*/*Fm*, the result is found to be a consistent value of roughly 0.83.	It represents the photochemical efficiency of PS-II and is used as an indicator of stress or damage to the photosynthetic system.
	*Φ_PSII_*	*Φ_PSII_ = (F_m_′ − F′)*/*F_m_′*, this parameter doesn’t need a dark-adapted measurement, so it is a commonly measured light-adapted parameter.	It represents the operating efficiency of PS-II photochemistry.
	*F_q_’*/*F_v_’*	*F_q_′*/*F_v_′ = (F_m_′ − F′)*/*F_v’_*, this parameter also doesn’t need a dark-adapted measurement.	It reflects the level of photoprotective quenching of fluorescence, and indicates the onset of photoinhibition.
	*NPQ*	*NPQ = (F_m_ − F_m_′)*/*F_m_′*, also be calculated as *(F_m_*/*F_m_′*) − 1.	*NPQ* is the non-photochemical quenching coefficient, which evaluates the rate constant for heat loss from PS-II.

PS-II: Photosystem II.

## Data Availability

Not applicable.

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
