# Peer review of "A Synthetic Review of Various Dimensions of Non-Destructive Plant Stress Phenotyping"

_plants, 2023, doi:10.3390/plants12081698_

Round 1

Reviewer 1 Report

Dear editorial team and authors,

 I have reviewed the manuscript entitled "A Synthetic Review of Various Dimensions of Non-Destructive Plant Stress Phenotyping" and have some comments to share. The manuscript contains interesting content, but it appears that the authors have spent a significant amount of time and effort collecting information associated with physiological parameters and regions where spectra can be informative. However, this information is not necessarily associated with the determination of stress in plants. While there is a significant body of work presented, much work needs to be done to make this understandable and publishable material. Therefore, I recommend that this manuscript should be published under revision and incorporated with a few points to make it clearer.

 General comments:

1.      The authors use the term "Non-Destructive Plant Stress Phenotyping" throughout the manuscript, but it is not clear how this term relates to the main idea of their work or the determination of stress in plants. The title of the manuscript seems to be used as an advertising strategy to attract readers' attention.

2.      While the manuscript is highly relevant to phenotyping, it is not clear how the tools presented can be applied to the intended message of "Non-Destructive Plant Stress Phenotyping." The message is not transmitted clearly, and the authors spend a large amount of text explaining aspects that are not relevant to the theme. The authors should emphasize the different types of stress that can be detected and provide examples of current applications

3.      The authors use terminology in an unconfident manner, which leads to doubts about the firm background and main idea of the manuscript. I suggest that the authors review recent literature to make their work more convincing, and reproducible, and provide a bigger contribution to science.

4.      The manuscript lacks definitions or descriptions of main ideas related to digital agriculture, agriculture 4.0, smart agriculture, data science big data, and artificial intelligence, among others. I suggest that the authors provide a comparison or clearer description of these ideas, including advantages, disadvantages, limitations, etc. It is necessary to explain how these technologies can be integrated into decision support systems.

5.      The manuscript shows a limited understanding of the mechanisms involved in the development of the design and applicability of a device to the indirect measurement of “phenotyping ” variables in agriculture or biological science. I urge the authors to review relevant scientific literature and re-write the sections. This part looks more like a conclusion than a discussion.

6.      The manuscript lacks a clear connection between sections, making it challenging to follow the flow of the document. I suggest that the authors improve the relationship between sections to achieve a clearer connection.

Overall, the manuscript has potential, but the authors need to address the issues mentioned above to make it more readable and understandable.

Author Response

Comment 1: The authors use the term "Non-Destructive Plant Stress Phenotyping" throughout the manuscript, but it is not clear how this term relates to the main idea of their work or the determination of stress in plants. The title of the manuscript seems to be used as an advertising strategy to attract readers' attention.

Re: Thank you for your valuable feedback. But we have to explain that “Non-Destructive” is the sharing feature of all presented technologies (tools). This is also the reason that all these phenotyping tools, either 1D spectroscopy, 2D imaging or 3D phenotyping, are non-invasive optical sensing techniques. Thus, the term "Non-Destructive Plant Stress Phenotyping", used throughout our manuscript, is indeed our main theme. We apologize for any confusion caused by not making this clearer earlier, but we have rephrased these to better convey our ideas and be more relevant to the theme of non-destructive plant stress phenotyping.

Comment 2: While the manuscript is highly relevant to phenotyping, it is not clear how the tools presented can be applied to the intended message of "Non-Destructive Plant Stress Phenotyping." The message is not transmitted clearly, and the authors spend a large amount of text explaining aspects that are not relevant to the theme. The authors should emphasize the different types of stress that can be detected and provide examples of current applications.

Re: "Non-destructive Phenotyping", "Various Dimensions", and "Plant Stress" are the three main aspects of this manuscript. "Plant Stress" is the specific application scenario for the non-destructive phenotyping tools discussed in the manuscript. These tools are further categorized into various dimensions, such as 1D, 2D, and 3D. That is why we did not focus on the different types of stress but on their common parts. We have revised the text to better emphasize the common aspects, such as the spectral characteristics or stress parameters that can be tested using these methods. We believe these revisions would make the manuscript more clear and more informative.

Comment 3: The authors use terminology in an unconfident manner, which leads to doubts about the firm background and main idea of the manuscript. I suggest that the authors review recent literature to make their work more convincing, and reproducible, and provide a bigger contribution to science.3.

Re: Thank you for bringing to our attention that we used terminology in an unconfident manner, which we were unaware of. Although we are confident that our approach is reasonable and correct, the unconfident manner may lead to confusion and unreliability. Therefore, we have revised this text and will be more cautious in the future. It's essential to strike a balance between being circumspect and confident in our writing, as well as being open to allow for different perspectives and interpretations.

Comment 4: The manuscript lacks definitions or descriptions of main ideas related to digital agriculture, agriculture 4.0, smart agriculture, data science big data, and artificial intelligence, among others. I suggest that the authors provide a comparison or clearer description of these ideas, including advantages, disadvantages, limitations, etc. It is necessary to explain how these technologies can be integrated into decision support systems.

Re: We have recognized that these concepts could add insights to our manuscript, and therefore we have already expanded on them in the last "Conclusions " part . ( For example, we have added: The integration of artificial intelligence (AI) and big data has shown great potential to revolutionize plant stress phenotyping, with the ability to identify patterns and correlations through real-time, multi-dimensional monitoring. This technology could enable the prediction and management of plant stress and facilitate plant breeding programs...)

Comment 5: The manuscript shows a limited understanding of the mechanisms involved in the development of the design and applicability of a device to the indirect measurement of “phenotyping” variables in agriculture or biological science. I urge the authors to review relevant scientific literature and rewrite the sections. This part looks more like a conclusion than a discussion.

Re: We have revised and also find it meaningful to divide our former section ("Conclusion and Perspectives") into two separate sections ( "Section 5: Discussion" and “Section 6 Conclusion”). which has made our article more organized and more coherent. We also analyze and summarize various dimensions of plant stress phenotyping techniques, all of which are indirect optical sensing methods, that have been used in various dimensions of plant stress phenotyping.

Comment 6: The manuscript lacks a clear connection between sections, making it challenging to follow the flow of the document. I suggest that the authors improve the relationship between sections to achieve a clearer connection.

Re: The manuscript is organized according to ordered spatial dimensions (1D, 2D, 3D). Within each dimension, we introduced various non-destructive phenotyping techniques (Vis-IR and ChlF 1D spectroscopy, Visible, ChlF, Hyperspectral, IR 2D imaging, Lidar, PET…3D phenotyping), along with their corresponding data acquisition and data analysis methods. We then explored multi-dimension. Although the sections may appear separate at first glance, they share a common feature, namely non-destructive optical plant stress phenotyping methods. We have updated Figure 1 and the instructions to provide better clarity and to emphasize the connection between the different sections. We believe that these revisions will enhance the cohesiveness and comprehensibility of the manuscript. For example: [ The article is divided into six sections, with the first introducing the background information and setting the context, emphasizing the importance of a synthesis review for various dimensions of phenotyping. Sections 2-4 cover phenotyping of 1D, 2D, and 3D, respectively, including data-acquiring technologies and corresponding data-analyzing methods. Section 5 analyzes the advantages and disadvantages of different plant stress phenotyping dimensions, and proposes trends for multi-dimensional phenotyping, namely combining spatial, spectral, and temporal dimensions. The conclusion summarizes various dimensions of plant stress phenotyping techniques, and emphasizes the need for an appropriate blend of these dimensions for precise and timely stress detection.] We hope that these amendments will meet your approval and we look forward to your continued feedback. 

Reviewer 2 Report

This paper discuss about the plant stress phenotyping. However, this review needs some necessary modifications.

1. Method description is ok, however, application of these methods are missing. 

2. How these methods are utilized and help scientists to make some reasonable conclusions is not explained in this paper.

3. Discussion on advantages and disadvantages of all these techniques should be discussed.

4. What are the advantages and disadvantages of different phenotypic methods should also be discussed?

5. Integration of deep learning and other techniques and their application should be included.

Author Response

Comment 1 & 2: Method description is ok, however, application of these methods are missing. How these methods are utilized and help scientists to make some reasonable conclusions is not explained in this paper.

Re: Thank you for your valuable feedback. We have revised our manuscript as your suggestion, and the necessary description of the application of these methods are added. List some for example.:

 (1) By applying appropriate spectral analysis, the changes in the reflectance signal can be extracted to characterize the plant's phenotype and even assess plant-specific genotype responses to biotic and abiotic stress [43-45]. For instance, Matthew [45] dis-cusses the use of reflectance spectroscopy for the early detection of plant physiological responses to different levels of water stress.

(2) S.P. Adav et al. utilized digital images of the potato leaves for estimating the chlorophyll content. And M. Janani describes a method for estimating the nitrogen nutrient range in groundnut crops using RGB images and CNN-based HVN model. These methods can be useful for high-throughput plant phenotyping. 

(3) Martínez.[92] presents a methodology for the early detection of fungal infections by measuring the temperature changes in grapes using infrared thermography.

(4) Anna Kicherer [87] utilized RGB and NIR imaging to analyze grapevines under drought stress and applied machine learning to identify patterns associated with drought tolerance. The study showed that RGB and NIR imaging can accurately differentiate between drought-tolerant and drought-sensitive grapevine varieties.

(5) The author in [138] notes that PET imaging has been successful in investigating the dynamic transport of nutrients, phytohormones, and photoassimilates.

Comment 3 & 4: Discussion on advantages and disadvantages of all these techniques should be discussed. What are the advantages and disadvantages of different phenotypic methods should also be discussed?

Re: As your suggestion, those discussions have been expanded. For example: (In summary, each has its advantages and disadvantages. 1D spectroscopy may gradually be less used, for limited spatial information. But the 1D spectral analysis is the basis and reference for higher dimension image processing, and it occupies less data storage while rich in spectral information. 2D imaging technologies are the mainstream, thanks to advancements in imaging sensors, computational devices, and the availability of a wide range of processing algorithms. 3D imaging and multi-dimension are important contemporary and future trends, capable of providing an excellent solution for comprehensive stress identification and prediction, while collecting various images and processing vast data is a practical issue. Supported by the development of high-resolution compact portable data acquiring devices, and high-performance computational data-processing algorithms, multi-dimension phenotyping will drive new research in agriculture), and (As for plant stress phenotyping, the surface traits of 3D models may not be fine enough to catch nuance changes in plants under stress. Due to limitations in point cloud density, surfaces are often incomplete or sparse. For this reason, multi-view 2D imaging is often better suited for capturing surface information. However, with the development of Light Detection and Ranging (LiDAR), high-resolution 3D models are becoming increasingly promising), and (However, thermal-IR imaging is vulnerable to environmental changes. Such as meteorological conditions and crosstalk of other emitted and reflected thermal radiation sources. For this reason, calibration and ground data collection are necessary. Meanwhile, data processing needs to be carried out for correct temperature retrieval). Meanwhile, we also add a discussion part to analyze these techniques, which would help our work to be more informative.

Comment 5: Integration of deep learning and other techniques and their application should be included.

Re: we have revised and included the information about the integration of deep learning and other techniques, as well as their application. For example: Li [118] use a U-Net architecture, which is a popular CNN architecture for image segmentation tasks, and modified the U-Net mode to take the point cloud data as input, then output a segmentation mask that identifies the different plant organs. We hope that these amendments will meet your approval and we look forward to your continued feedback.

Reviewer 3 Report

Review

A Synthetic Review of Various Dimensions of Non-Destructive 2

Plant Stress Phenotyping 3 Dapeng Ye1,2, Libin Wu1,2, Xiaobin Li1,2, Tolulope Opeyemi Atoba11,2, Wenhao Wu1,2 and Haiyong Weng 1,2*.

General comments:

The article is very valuable and interesting, it provides an overview of non destructive methods on plant phenotyping in the premise for stress recognition for agriculture and environmental protection. The paper adopts the correct division of methods based on the dimensionality of the data used for stress detection: 1D, 2D and 3D, including spatial and spatio-temporal data. In addition, a pipeline diagram of data processing and analysis is included in the form of diagrams. The review, given the enormity of the problem, intended to be synthetic, covers the vast majority of methods used in non-destructive plant phenotyping: chlorophyll fluorescence, various types of spectroscopy,visible-light imaging, FR/NIR, multi and hyperspectral, Lidar, CT, PET, MRI. The advantage and the most interesting part of the article is the presentation of modern methods of Machine Learning: traditional and Deep learning. The whole is supplemented with a rich literature on the subject (158 items).

The authors did not avoid many shortcuts, simplifications, especially in the introductory part of the work. It would be necessary to further define the area of application of the presented methods from a biological perspective, for example, to present what processes we can study with the help of chlorophyll fluorescence ?.  

I agree that PAM fluorescence gives us more information about the status of PSII. It also gives an opportunity for more advanced experimental analysis. However it gives with const of time, since the full PAM analysis of one point on the plant leave takes from 15 to 45 min. So, fast chlorophyll fluorescence (1-10 s), especially on detached leaves provide us with better results in terms of physiological reaction of plants to environmental stresses

The article needs to be rewritten (please consult with native English speaker), in terms of use of colloquial expressions, abbreviations, etc.

Specific comments are provided below:

Lines 24-39 - This part of the text needs extensive rewriting and supplementation. The study of environmental stress in plants and its detection has much greater importance and wider applications than stated in the text. 

Please consult for example:

(1) Russian Journal of Plant Physiology, 2016, Vol. 63, No. 6, pp. 869-893

(2) Kalaji et al. 2014, 2017 "Frequently asked questions about chlorophyll fluorescence..."

Lines 171-173 - SSPS, Excel, etc., are not statistical methods but software. Please rewrite the sentence: ".....with/using SSPS, Excel, Matlab software"

Lines 187 There are many inaccuracies in the text and abbreviated expressions

 Shoul be "plant physiological status"

Lines 203 should be "....photochemical quenching"

Figure 1 needs readability improvement (invisible details in the figure)

Table 1 Description of ChlF parameters are somewhat odd, they are please consult the literature

Lines 239 Machine vision is not equall.... Straighforward, Please, re

Translated with www.DeepL.com/Translator (free version)

Author Response

Specific comments: Lines 24-39 - This part of the text needs extensive rewriting and supplementation. The study of environmental stress in plants and its detection has much greater importance and wider applications than stated in the text. Please consult for example:(1) Russian Journal of Plant Physiology, 2016, Vol. 63, No. 6, pp. 869-893 (2) Kalaji et al. 2014, 2017 "Frequently asked questions about chlorophyll fluorescence..."

Re: Thank you for your valuable feedback. We agree that we have simplified the introduction of “the study of environmental stress in plants…”, we have revised this section to emphasize the importance of studying plant stress and its detection, while also improving the introduction's clarity and consistency. [This part has been revised as: Unfavorable factors that affect the metabolism, growth or development of plants are known as stressors. Plants have developed a variety of adaptive strategies to deal with environmental stressors, enabling them to survive and even thrive in unfavorable conditions. These observable traits, which arise from the interaction between a plant's genotypes and the environment, are the phenotypes we aim to obtain. Understanding these stress-induced phenotypes can aid plant breeders in developing stress-tolerant plant varieties, and can inform the development of management strategies to mitigate the effects of environmental stress on plants. This is crucial for ensuring food safety and ecosystem conservation. To accomplish this, a comprehensive understanding of plant stress phenotyping techniques is necessary…]. Furthermore, we have consulted the literature suggested by you, such as the Russian Journal of Plant Physiology [1] and Kalaji et al.[2, 3] to enhance our discussion, which are rather helpful.

Lines 171-173 - SSPS, Excel, etc., are not statistical methods but software. Please rewrite the sentence: ".....with/using SSPS, Excel, Matlab software"

Lines 187 There are many inaccuracies in the text and abbreviated expressions shoul be "plant physiological status"

Lines 203 should be "....photochemical quenching"

Lines 239 Machine vision is not equall.... Straighforward, Please, re Translated with www.DeepL.com/Translator (free version)

Re: We have corrected the inaccuracies in the text in the listed lines above. And we also carefully checked the English in our manuscript and have made extensive improvements to ensure that our work is accurately represented and effectively communicated to your readers.

Figure 1 needs readability improvement (invisible details in the figure).

Re: For Figure 1, we have redrawn the diagram to improve clarity and readability. Additionally, we have improved the quality of Figure 2, 3,4 as well.

Table 1 Description of ChlF parameters are somewhat odd, they are please consult the literature.

Re: Regarding Table 1, we have revised and expanded the table, adding one column to describe the parameters, and the former one is more related to measurement. These changes are marked in grey color as the following table 1 shows. As for the terms "primary parameters" and "deduced parameters" used in the "Description of ChlF parameters" section, also can be seen used in the literature (Van Kooten. The use of chlorophyll fluorescence nomenclature in plant stress physiology[J]. Photosynthesis research, 1990; and Papageorgiou. Chlorophyll a fluorescence: a signature of photosynthesis. Dordrecht: Springer Netherlands,2004, and among others [4-7]).

Parameter

Measurement & Calculation

Description

Preliminary parameters

Fo

Switch on the measuring light, and get the parameter Fo.

Minimal fluorescence of chlorophyll a in dark-adapted leaves, indicating the baseline fluorescence of the sample.

Fm

Offer a pulse, then induces Fm.

Maximal fluorescence of chlorophyll a in dark-adapted leaves.

F'

Switch on actinic light, followed by an initial rise in fluorescence. Then fluorescence quenches due to the increasing competition with photochemical and non-photochemical events. This state is also named the light-adapted state.

F’ represents the ChlF yield in the light-adapted state in the presence of actinic light.

Fm'

Offer a pulse to the light-adapted state, and get the parameter Fm'.

Maximal fluorescence of chlorophyll a in light-adapted leaves.

Fo'

Switch off the actinic light and measure immediately, recording Fo'. But the accurate measurement is complex; an alternative approach is to calculate Fo'.

Minimal fluorescence of chlorophyll a in light-adapted leaves.

Deduced parameters

Fv 

Fv =Fm-Fo, the difference between Fm and Fo is the variable fluorescence Fv.

It is related to the maximum quantum yield of PS-II, reflecting the amount of chlorophyll molecules that are in the open reaction centers and actively.

Fv/Fm

Fv/Fm =(Fm-Fo)/Fm, the result is found to be a consistent value of roughly 0.83. 

It represents the photochemical efficiency of PS-II and is used as an indicator of stress or damage to the photosynthetic system.

ΦPSII

ΦPSII = (Fm’-F')/Fm’, this parameter doesn’t need a dark-adapted measurement, so it is a commonly measured light-adapted parameter.

It represents the operating efficiency of PS-II photochemistry.

Fq'/Fv'

Fq'/Fv' = (Fm’-F')/Fv', this parameter also doesn’t need a dark-adapted measurement.  

It reflects the level of photoprotective quenching of fluorescence, and indicates the onset of photoinhibition.

NPQ

NPQ = (Fm-Fm′)/Fm′, also be calculated as (Fm/Fm')-1.

NPQ is the non-photochemical quenching coefficient, which evaluates the rate constant for heat loss from PS-II.

General comments: It would be necessary to further define the area of application of the presented methods from a biological perspective, for example, to present what processes we can study with the help of chlorophyll fluorescence?

Re: We agree that it is important to define the biological processes that can be studied using the presented methods. For instance, chlorophyll fluorescence is a valuable tool for investigating various photosynthesis-related processes in plants, such as light harvesting, energy transfer, electron transport, and carbon fixation. By analyzing the corresponding fluorescence parameters, different aspects of these processes, including their efficiency and regulation, can be assessed. Therefore, we have expanded on the details of using spectral characteristics, not only for ChlF but also other optical phenotyping methods, to evaluate plant stress responses and monitor plant growth and development in our manuscript.

PAM fluorescence gives us more information about the status of PSII, and gives an opportunity for more advanced experimental analysis. However it gives with const of time, since the full PAM analysis of one point on the plant leave takes from 15 to 45 min. So, fast chlorophyll fluorescence (1-10 s), especially on detached leaves provide us with better results in terms of physiological reaction of plants to environmental stresses. Provided more detail about fast chlorophyll fluorescence (1-10 s).

Re: These have been expanded in our manuscript to complete the ChlF methods, for example: [There are also alternative methods, namely OJIP-test or fast (prompt) ChlF. OJIP means a fast (often fewer than 1s) increasing phase of chlorophyll a fluorescence induced by constant light excitation. O is the origin (the minimum fluorescence), J and I are in the intermediate levels, and P is the peak. One of the most typical testing methods is using PEA (plant efficient analysis). It can measure the fast fluorescence induction kinetics, which reflects the state of the electron transport chain in PS-II...]. 

The article needs to be rewritten in terms of use of colloquial expressions, abbreviations, etc.

Re: We have rewritten the article to ensure that colloquial expressions and abbreviations are used correctly. We hope that these amendments will meet your approval and we look forward to your continued feedback.

Round 2

Reviewer 2 Report

Authors addressed my and other reviewer's concerns appropriately. Manuscript is in good shape for publication.

Reviewer 3 Report

.